# Malaria Prevention for Pregnant Women and Under-Five Children in 10 Sub-Saharan Africa Countries: Socioeconomic and Temporal Inequality Analysis

**DOI:** 10.3390/ijerph21121656

**Published:** 2024-12-11

**Authors:** Denis Okova, Akim Tafadzwa Lukwa, Robinson Oyando, Paidamoyo Bodzo, Plaxcedes Chiwire, Olufunke A. Alaba

**Affiliations:** 1Health Economics Unit, School of Public Health, Faculty of Health Sciences, University of Cape Town, Anzio Road, Observatory 7925, South Africa; tafadzwalukwa@gmail.com (A.T.L.); bodzo.panashe@gmail.com (P.B.); olufunke.alaba@uct.ac.za (O.A.A.); 2Health Economics Research Unit, KEMRI-Wellcome Trust Research Programme, Nairobi P.O. Box 43640-00100, Kenya; royando@kemri-wellcome.org; 3Western Cape Department: Health, Western Cape Province, P.O. Box 2060, Cape Town 8000, South Africa; plaxcedes.chiwire@healtheconomics.co.za; 4Department of Health Services Research, CAPHRI Care and Public Health Research Institute, Maastricht University, 6200 MD Maastricht, The Netherlands

**Keywords:** child health, health equity, Insecticide Treated Nets (ITNs), Intermittent Preventive Treatment (IPTp), maternal health, socioeconomic disparities

## Abstract

Background: Malaria remains a public health challenge in low- and middle-income countries (LMICs). Despite gains from strategies like Insecticide-Treated Nets (ITNs) and Intermittent Preventive Treatment during pregnancy (IPTp), significant socioeconomic inequalities persist, particularly among pregnant women and children under five. This study analyzed temporal and socioeconomic inequalities in malaria prevention in sub-Saharan Africa (SSA). Methods: Nationally representative Demographic Health Surveys from 10 SSA countries (Mozambique, Burkina Faso, Tanzania, Côte d’Ivoire, Madagascar Kenya, Rwanda, Nigeria, Uganda, and Cameroon) were used, comparing two time periods. Changes in ITN use by pregnant women and children under five, as well as IPTp coverage, were assessed. Inequalities based on socioeconomic status (SES) and residence were analyzed using the Erreygers Normalized Concentration Index and Theil index. Results: The results revealed significant variability in ITN use and IPTp coverage within countries. Eight countries showed improvements in ITN use during pregnancy, with Nigeria seeing a 173.9% increase over five years. Burkina Faso and Tanzania consistently reported high ITN use (~87%) in children under five. IPTp coverage increased in all countries except Kenya. Decomposition using the Theil index indicated that within-group inequalities, particularly based on SES and residence, were the primary drivers of disparities. Conclusions: To ensure progress toward universal health coverage, malaria prevention programs must prioritize vulnerable populations and be continuously evaluated.

## 1. Introduction

Malaria remains one of the most common diseases in low-and middle-income countries (LMICs). In 2022, the World Health Organisation (WHO) reported that there were an estimated 249 million malaria cases and 608,000 malaria deaths globally, most of which occurred in LMICs [1]. The burden of malaria is most felt in the African region with 94% of all global malaria cases occurring in sub-Saharan Africa (SSA), and four countries, Nigeria (27%), the Democratic Republic of the Congo (12%), Uganda (5%), and Mozambique (4%), accounting for almost half of all cases globally in 2022 [1]. Vulnerable populations such as pregnant women and children under the age of five in malaria-endemic countries face a heightened risk of malaria. In 2022 alone, 12.7 million malaria cases were reported among pregnant women across 33 SSA countries while a significant proportion of malaria deaths occurred among children under five years [1,2]. In the past two decades, however, significant gains have been made in the fight against malaria [3]. For instance, between 2000 and 2021, the global malaria mortality decreased by 32.6% while the global malaria incidence decreased by 21.1% [4]. These gains have been made possible by an increased coverage of malaria interventions such as insecticide-treated nets (ITNs), in-door residual spraying (IRS), malaria treatment, and chemoprevention [1].

Prior to 2008, malaria control interventions were targeted at pregnant women and children aged under five by prioritizing the distribution of ITNs through routine vaccinations, antenatal clinics, and the social marketing of subsidized or free malaria commodities [5,6]. Equity in the distribution of malaria interventions distributed through these channels emerged as a concern as wealthy urban households who had better access to healthcare services relative to rural-poor households enjoyed better coverage by these malaria interventions [7,8]. This approach to the distribution of malaria interventions changed with the call from the United Nations Secretary General for universal coverage of malaria interventions [9]. Consequently, in malaria-endemic countries, there was a shift from targeted to mass ITN distribution through community delivery supported by routine services [3]. In addition, in 2004, in response to emerging parasite resistance, fast-acting artemisinin-based combination therapies (ACTs) replaced monotherapy as the recommended first-line treatment for uncomplicated malaria [10]. Through the support of the Global Fund and novel financing mechanisms such as the Affordable Medicines Facility-malaria, ACTs were made available in malaria-endemic countries [11]. The involvement of community health workers, among other innovative distributional channels in malaria-endemic countries, reduced the inequality gap between the poorest and least poor in the coverage of these malaria interventions [12].

Intermittent preventive treatment during pregnancy (IPTp), the use of ITNs, and the effective case management of malaria and anaemia are the package of interventions recommended by the WHO in areas with the stable to high transmission of *Plasmodium falciparum* to tackle malaria in pregnancy [13]. Sulphadoxine-pyrimethamine (SP) is currently the most effective drug for IPTp in areas of high or seasonal transmission as it has been shown to lower placental infection rates and increase both maternal haemoglobin levels and the infants’ birth weight [14,15]. Intermittent preventive therapy with SP (IPTp-SP) for malaria in pregnancy is delivered to pregnant women as they attend antenatal clinics (ANC) [16]. Evidence suggests that despite high ANC attendance in most SSA countries, coverage with ITNs and IPTp-SP remains sub-optimally below the 80% target [17]. Evidence from meta-analyses of factors affecting the delivery, access, and use of interventions to prevent malaria in pregnancy in SSA have identified socioeconomic status [18,19,20], education, knowledge, and perceptions about malaria [21,22], the number and timing of ANC visits [20,22], the number of pregnancies, and health system inadequacies [20,21] as factors that influence intervention coverage as well as access to ANC services. 

Whilst available evidence suggests that progress has been made in malaria prevention, there are persistent inequalities in the uptake of malaria prevention strategies between the poorest and the richest [23,24]. Previous studies have evaluated inequalities in the coverage of malaria control interventions, mostly ITNs, and outlined the role of socioeconomic status as one of the important predictors in the use and coverage of malaria control interventions. While ITN ownership has become pro-poor as a result of mass net distribution programs, net utilisation is largely pro-rich in many contexts, highlighting persistent disparities in the effective use of this key prevention strategy [25,26]. Two previous studies assessed equity in the coverage of malaria control interventions across a number of malaria-endemic African countries [3,7]. These studies provided mixed equity findings on the coverage of malaria control interventions. A dated study by Steketee et al. found that in 13 out of the 25 countries assessed, malaria treatment for febrile children and IPTp-SP for pregnant women favoured the rich whilst ITN ownership favoured poor households [7]. Conversely, a study by Galactionova, Smith, de Savigny, and Penny [3], which examined data from 30 malaria-endemic sub-Saharan African (SSA) countries using Demographic and Health and Malaria Indicator Survey data between 2005 and 2015, found that there were still wide inequities in the coverage of malaria interventions and their distribution by socioeconomic status within and across countries; the odds are still stacked against the poor [3]. 

To our knowledge, previous studies examining inequalities in the utilization of malaria prevention interventions among pregnant women and children under five have largely focused on between-group disparities, often classifying populations by socioeconomic status or rural–urban residence. While these approaches highlight differences across broad categories, they tend to overlook within-group inequalities, which account for the diversity and heterogeneity within these groups. For instance, rural populations may experience varying levels of access to malaria prevention tools based on geographic remoteness or cultural factors, while urban populations may face inequities linked to informal settlements or overcrowding. Failing to consider both between- and within-group inequalities may obscure critical inequities and hidden vulnerabilities, particularly among the most marginalized individuals in both rural and urban settings. This oversight limits the effectiveness of public interventions and their ability to address the diverse needs of these populations comprehensively. Table 1 below provides a summary of between-group and within-group inequalities. Utilizing the most recent Demographic Health Survey (DHS) datasets from 10 SSA countries, the present study adds to existing evidence by (1) analysing the within-country trends in the uptake of malaria prevention interventions (ITN use by pregnant women and children under 5, IPTp coverage in pregnancy) over time, (2) investigating the within-country between-socioeconomic and residence group intertemporal inequalities in the uptake of these malaria prevention strategies, (3) examining the intra-country within-socioeconomic and residence group intertemporal inequalities in the uptake of these malaria prevention strategies, and, finally, examining the contribution of between- (socioeconomic and residence status) and within- (socioeconomic and residence status) group inequality to the overall inequality in ITN use and IPTp coverage. This study focused on analysing within-country trends rather than direct cross-country comparisons, as the base and end years varied across countries. This variation in timeframes could introduce inconsistencies, making direct comparisons between countries less meaningful.

By understanding trends in the uptake of malaria prevention strategies as well as characterizing associated between- and within-group inequalities, this study can support evidence-based policy aimed at achieving equitable access to malaria prevention tools. Also, it can help align national malaria programs with global health goals, such as Universal Health Coverage (UHC), Sustainable Development Goal 3 (Good Health and Well-being), and Sustainable Development Goal 10 (Reduced inequalities). Additionally, focusing on both pregnant women and children under five in inequality studies enables the development of targeted public health interventions that address the unique needs of these highly vulnerable populations. These groups not only share a heightened susceptibility to malaria but also have significant public health implications associated with their infections. Moreover, both groups bear a disproportionate burden of malaria, a leading cause of morbidity and mortality in sub-Saharan Africa, highlighting the urgency of addressing the inequities that limit their access to effective prevention and treatment. This research is also particularly timely as global progress in malaria control has stalled in recent years, as highlighted by the WHO, signaling the urgent need to rethink “business as usual” approaches [1]. Moreover, recent health emergencies and climate change are jeopardizing hard-won gains, underscoring the importance of renewed efforts [1]. This work is also critical as countries scale up malaria vaccination programs [1], presenting a pivotal opportunity to enhance and sustain progress.

## 2. Materials and Methods

### 2.1. Data

This study utilized publicly available nationally representative data from the Demographic Health Surveys (DHS). The countries studied were Mozambique (2011 and 2022/23), Burkina Faso (2010 and 2021), Tanzania (2015/16 and 2022), Côte d’Ivoire (2011/12 and 2021), Madagascar (2008/9 and 2021), Kenya (2014 and 2022), Rwanda (2014/15 and 2019/20), Nigeria (2013 and 2018), Uganda (2011 and 2016), and Cameroon (2011 and 2018). See Figure 1 for a visual representation of these countries of interest. These countries were chosen based on two criteria: firstly, the availability of recent datasets, and secondly, having the highest malaria incidence as reported by the 2023 World Health Organization Malaria report [1]. Cross-country comparisons may be subject to bias due to variations in base and end years; however, this was not the objective of this study. Instead, the focus was on examining intra-country trends. With the recommendation from DHS [27], the household member recode (PR), the individual recode file for women (IR), and the household recode (HR) file were merged for each country and each year, which resulted in 20 datasets that were appended into one file for final analysis. 

### 2.2. Ethical Considerations

The secondary analysis in this study utilized publicly available Demographic and Health Survey (DHS) data from 10 sub-Saharan African countries. As such, no direct ethical approval was sought for this analysis. The original DHS data collection protocols were reviewed and approved by the respective national ethics review boards and the Institutional Review Board (IRB) of ICF International, ensuring the protection of participants’ welfare and rights. The data are de-identified and publicly accessible, minimizing any risk to participants. Additionally, the data are governed by strict confidentiality protocols in accordance with the DHS Program’s data-use policies.

### 2.3. Study Variables

#### 2.3.1. Outcome Variables

The outcome variables for this study were as follows: ITN use among pregnant women at the time of the survey, ITN use among children under five, and IPTp coverage among women who had a live birth or stillbirth within two years preceding the survey. These variables were operationalized based on DHS guidance and their definition in previous studies as well as by virtue of being malaria-intervention-coverage indicators [1]. ITN use by pregnant women was defined as women aged 15–49 who were pregnant at the time of the survey, lived in a household with at least one insecticide-treated net (ITN), and slept under an ITN the night before the survey [27,28]. The use of insecticide-treated nets (ITNs) by children under five was described as children living in households with at least one ITN who slept under it on the night prior to the survey [29]. The third variable, IPTp coverage, was defined as women aged 15 to 49 who had a live birth or stillbirth within two years before the survey and received three or more doses of Sulphadoxine-Pyrimethamine (SP) during their pregnancy [30,31].

For all countries, all these three outcome variables were computed, except for Rwanda where it was not possible to compute IPTp coverage due to data limitations. Regardless, Rwanda was included as it is one of the countries where malaria incidence has significantly declined in recent years. In 2022, the incidence rate was reduced to 76 cases per 1000 people from 409 per 1000 in 2016, marking a 39% decrease [1].

#### 2.3.2. SES and Place of Residence

The socioeconomic status (SES) and place of residence are key determinants that help in understanding disparities across different population groups. This study measured SES using the already computed wealth index in the DHS. Place of residence was determined using rural versus urban living.

### 2.4. Statistical Analysis

Data analysis was carried out using STATA 17. First, preliminary analysis was conducted to describe the sample and explore patterns of ITN use in pregnancy and under-fives as well as IPTp coverage. Consequently, Erreygers Normalized Concentration indices (ENCI) and Theil indices were computed to understand inequalities in the outcome measures. All the findings of this study ought to be interpreted bearing in mind the differences in time periods between the two datasets for the various countries.

#### 2.4.1. Erreygers Normalised Concentration Index (ENCI)

Inequalities in ITN use and IPTp coverage were first computed using the ENCI, a single summary measure that ranges from −1 to +1 [32]. A positive ENCI denotes that a health variable (ITN use and IPTp coverage) is more concentrated among wealthier socioeconomic groups and vice versa. A pro-rich inequality (positive ENCI) means that higher socioeconomic groups are more likely to utilize or access the health service in question. A pro-poor inequality (negative ENCI), on the other hand, means access to or utilization of health services, resources, or interventions is disproportionately concentrated among poorer individuals or socioeconomic groups. An ENCI of zero denotes perfect equality and the uniform distribution of malaria prevention resources across all individuals. 

The Erreygers normalized concentration index is calculated as shown below:$$E\left(C\right)=\frac{4\mu }{b-a}C$$
where

*μ* is the mean of the health variable (ITN use in under-fives, ITN use in pregnancy, IPTp coverage).

*b* and *a* are the upper and lower bounds of the health variable, respectively.

*C* is the traditional concentration index.

The traditional concentration index is calculated as shown below: $$C=\frac{2}{\mu }\;Cov\;(y,R)$$
where

*y* is the health variable (ITN use in under-fives, ITN use in pregnancy, IPTp coverage).

*μ* is the mean of *y*

R is the fractional rank of individuals in the distribution of SES.

Cov denotes the covariance between *y* and *R*.

#### 2.4.2. Theil Indices

Given that the ENCI was unable to demonstrate the extent to which inequality in ITN use in under-fives, ITN use in pregnancy, or IPTp coverage is due to differences between and within SES and place of residence, this study computed Theil indices. This index, a Generalized Entropy index GE (2), permits the decomposition of inequality into its within-group and between-group components [33].

In this study, Theil indices were first calculated for the entire sample and then disaggregated by socioeconomic status (SES) and place of residence. Subsequently, the Theil indices were further decomposed to evaluate the contributions of both within-group and between-group components to the total inequality.

The GE (2) index ranges from 0 to infinity, where a value of 0 indicates perfect equality, and values above 0 signify increasing levels of inequality.

The GE (2) index is calculated as follows:$$GE\left(2\right)=\frac{1}{2}\sum_{i-1}^{n}({\frac{y_{i}}{\mu })}^{2}-1$$
where

$y_{i}$ is the outcome for individual i,μ is the mean of the outcome variable,

n is the total number of individuals.

The overall GE (2) index is then decomposed into within-group and between-group components as shown:$$GE\left(2\right)={GE}_{w}\left(2\right)+{GE}_{B}(2)$$
where

${GE}_{w}\left(2\right)$ represents within-group inequality and

${GE}_{B}(2)$ represents between-group inequality.

The within-group and between-group contributions to overall inequality in health outcomes (ITN use in under-fives, ITN use in pregnancy, IPTp coverage) were calculated as follows:$$Within-group\;contribution=\left(\frac{{GE}_{w}(2)}{GE(2)}\right)*100$$
$$Between-group\;contribution=\left(\frac{{GE}_{B}(2)}{GE(2)}\right)*100$$

## 3. Results

### 3.1. Descriptive Statistics

Table 2 presents the proportions of ITN use in pregnancy, ITN use by under-fives, and IPTp coverage. The results demonstrated a general trend of improvement in ITN use during pregnancy over time in most countries, although there were notable exceptions such as Kenya and Rwanda where slight decreases were observed. Nigeria saw a dramatic increase in ITN use during pregnancy from 29.82% in 2013 to 81.67% in 2018, and such a huge increase was also observed in Cameroon and Côte d’Ivoire indicating a strong push for malaria prevention. For ITN use by under-fives, Nigeria registered a low prevalence (31.48% in 2013), but this dramatically improved to 80% in 2018. Cameroon also demonstrated strong improvements over time, with more than a 20-percentage-point increase in under-five ITN use. Burkina Faso and Tanzania both showed consistently high levels of ITN use by under-fives, with both countries achieving approximately 87% by 2021/2022.

The prevalence of IPTp coverage has shown notable variations over time within individual countries. For instance, Burkina Faso demonstrated a steady increase in IPTp coverage, rising from 40.95% in 2010 to 56.68% in 2021. Similarly, Madagascar experienced a significant improvement, with IPTp coverage increasing from 1.93% in 2008/09 to 30.96% in 2021. In contrast, Kenya saw a decline in IPTp coverage, dropping from 10.55% in 2014 to 6.45% in 2022, indicating potential challenges in sustaining malaria prevention efforts during pregnancy.

In general, Burkina Faso consistently showed high and improving prevalences in all three areas (ITN use in pregnancy, ITN use by under-fives, and IPTp coverage), positioning it as a strong performer in malaria prevention. Nigeria also exhibited a remarkable improvement, particularly in ITN use in pregnancy and among under-fives, with significant increases between 2013 and 2018. However, Kenya and Rwanda showed slight declines or stagnation in key indicators, signalling potential areas for concern and further investigation into barriers to prevention efforts.

### 3.2. Concentration Indices and Curves

As shown in Table 3, in Côte d’Ivoire, ITN use during pregnancy among poorer populations increased over time, as reflected in the ENCI value, which decreased from −0.117 in 2011/12 to −0.216 in 2021. Nigeria observed a notable shift from pro-rich inequality (0.067 in 2013) to significant pro-poor inequality in 2018 (−0.156, *p* = 0.000), suggesting a redistribution of ITN use towards poorer populations. Rwanda and Kenya, on the other hand, show persistent pro-rich inequality in ITN use during pregnancy, with Rwanda’s inequality increasing slightly over time.

As shown in Table 4, there were general widening inequalities in IPTp coverage to the benefit of wealthier women across most countries. Cameroon, Mozambique, Nigeria, Côte d’Ivoire, and Madagascar had an increase in ENCI over two study periods, indicating a growing concentration of IPTp coverage among wealthier women.

From Table 5, Côte d’Ivoire and Nigeria experienced statistically significant shifts toward pro-poor outcomes in ITN use among children under five. For Côte d’Ivoire, the ENCI became more negative over time, shifting from −0.090 in 2011/12 to −0.183 in 2021, while for Nigeria, the ENCI moved dramatically from 0.060 (pro-rich inequality) in 2013 to −0.139 (pro-poor inequality) in 2018, reflecting a redistribution of ITN use to poorer households. Kenya, Mozambique, and Rwanda consistently exhibited pro-rich inequality, although in Kenya, this inequality reduced over time.

The Lorenz curves in Appendix A illustrate the distribution of ITN use in pregnancy, ITN use by under-fives, and IPTp coverage by SES across SSA in two different years per country. In general, the curves for IPTp coverage are well below the line of equality, indicating a pro-rich distribution. That is, of all the three interventions, IPTp coverage is mostly concentrated among wealthier households, pointing to the need to improve access to and utilization of IPTp in poor populations.

### 3.3. Theil Indices for ITN Use in Pregnancy and ITN Use by Under-Fives Distinguished by Place of Residence

Table 6 shows major improvements in equity for ITN use and IPTp coverage in Mozambique, Côte d’Ivoire, Nigeria, Uganda, and Cameroon, with substantial reductions in Theil indices over time, particularly for rural areas. 

In Mozambique, inequality in ITN use in pregnancy decreased significantly from 0.215 (urban) and 0.444 (rural) in 2011 to 0.164 (urban) and 0.221 (rural) in 2022/23, indicating reduced inequality across both urban and rural settings. For IPTp coverage, inequality increased dramatically, especially in rural areas, rising from 0.674 in 2011 to 1.874 in 2022/23, suggesting that rural women are now facing significantly more unequal access to IPTp services.

With ITN use by under-fives, inequality decreased over time, with the indices for urban areas moving from 0.236 to 0.099, and rural areas from 0.345 to 0.189 in 2022/23, indicating better distribution of ITN use among children. 

For Côte d’Ivoire, inequality in the use of ITNs in pregnancy decreased significantly from 0.429 (urban) and 0.295 (rural) in 2011/12 to 0.172 (urban) and 0.096 (rural) in 2021, showing much greater equity in ITN distribution by 2021. Similarly, for IPTp coverage, inequality fell drastically from 5.161 (urban) and 7.592 (rural) to 0.940 and 1.207, respectively, reflecting improved equity in IPTp coverage. With ITN use by under-fives, both urban and rural areas show reduced inequality, with urban GE(2) dropping from 0.407 to 0.186 and rural GE(2) from 0.285 to 0.090.

For Nigeria, inequality in ITN use in pregnancy significantly reduced, with urban GE (2) falling from 0.891 to 0.179 and rural GE (2) from 1.273 to 0.093 by 2018, indicating much greater equality in ITN use. IPTp coverage reported a dramatic decrease in inequality, with urban GE (2) dropping from 6.905 to 1.880 and rural GE (2) from 7.281 to 2.848, indicating a more equal distribution of IPTp coverage by 2018. Similarly, with ITN use by under-fives, inequality decreased sharply, with both urban and rural areas seeing significant improvements in access for both groups.

### 3.4. Theil Indices for ITN Use in Pregnancy and ITN Use by Under-Fives Distinguished by SES

Analyzing the Theil indices by SES (as shown in Table 7) shows an improved equity in ITN use in pregnancy, IPTp coverage, and ITN use by under-fives across several countries, particularly Mozambique, Nigeria, Burkina Faso, Tanzania, Côte d’Ivoire, and Cameroon, where inequality decreased significantly across the three socioeconomic classes. Of these countries, Nigeria demonstrated a dramatic improvement in ITN use and IPTp coverage for all groups, especially the poor, by 2018. Conversely, Kenya and Rwanda saw increases in inequality in malaria prevention, with poorer groups facing higher inequality. Of the three malaria prevention interventions, IPTp coverage generally exhibits the highest levels of inequality across most countries and years, but the trend shows improvement over time, especially among poor groups.

### 3.5. Decomposition of Theil Indices by Place of Residence

Countries recorded reductions in overall inequality in ITN use in pregnancy. Such countries include Cameroon (improved by 0.375 points), Côte d’Ivoire (improved by 0.208 points), and Mozambique (improved by 0.154 points). Conversely, Kenya and Rwanda recorded slight increases in overall inequality in ITN use in pregnancy (Table 8). For ITN use by under-fives, in Mozambique, Côte d’Ivoire, and Cameroon, the overall inequality reduced over time meaning that these countries made significant progress in reducing inequality in ITN access for this population. For IPTp coverage, Madagascar (21 points), Nigeria (4.74 points), and Mozambique (3.49 points) had high decreases in overall inequality suggesting improving equity in access to IPTp coverage. On the other hand, Kenya observed a notable increase in overall inequality for IPTp coverage, with an increase of 2.987 points. This indicates a widening gap in access to IPTp services, with advantaged groups increasingly benefiting more than disadvantaged groups by 2022.

Regarding the contribution of within- (rural or urban) and between- (rural versus urban) group components to the overall inequality, differences between rural versus urban living contributed very little to the overall inequality in all countries, with disparities between rural or urban populations being the dominant contributor of overall inequality across the board. 

Cameroon had a sizable increase in the between-group contribution for IPTp coverage (+1.74%), indicating widening gaps between people living in rural versus urban areas. Mozambique and Nigeria also saw increases in between-group contributions, though to a lesser extent. Burkina Faso had minimal between-group inequality in both years, indicating that most inequality in IPTp coverage is driven by within-group differences. Mozambique also reported a decrease in within-group inequality (−3.498 points), reflecting a large improvement in equity for IPTp coverage within groups.

Madagascar had a notable decrease in within-group inequality (−21.099 points), with almost all inequality coming from disparities within population groups (e.g., urban groups, rural groups). This signals a significant improvement in equity for IPTp coverage within the two groups over time. Lastly, Kenya experienced a worsening within-group inequality, with the within-group contribution rising by +2.985 points. This means that inequality worsened within people of the same place of residence, contributing to the overall increase in inequality for IPTp coverage.

### 3.6. Decomposition of Theil Indices by SES

Table 9 shows the decomposition of Theil indices by SES. It was observed that Cameroon had a decrease in overall inequality for ITN use in pregnancy (−0.375), while Rwanda had a slight increase in inequality for ITN use in pregnancy (+0.039). Nigeria saw a sizable decrease in overall inequality for ITN use among under-fives (−0.908), while Tanzania had a small decrease (−0.073), indicating less change over time. Madagascar had a large decrease in overall inequality for IPTp coverage (−21.099), showing a dramatic improvement in equity, while Kenya had a large increase in inequality for IPTp coverage (+2.987), indicating worsening access disparities.

Côte d’Ivoire reported a sizeable increase in between-socioeconomic-group inequality for ITN use in pregnancy, with the contribution from between-group disparities rising by 5.94%. This indicates that inequality between socioeconomic classes became more pronounced over time. Cameroon experienced a sizeable decrease in within-socioeconomic group inequality for ITN use in pregnancy, with the contribution of within-group disparities decreasing by 0.371 points. This suggests improved equity within socioeconomic groups, although overall inequality still remains.

On ITN use by under-fives, Cameroon demonstrated a large decrease in between-socioeconomic group inequality for ITN use by under-fives, with the contribution of between-group disparities reducing to 0% by 2018. This indicates a reduction in disparities between socioeconomic classes. Mozambique had a notable increase in between-socioeconomic group inequality for IPTp coverage, with the contribution from between-group disparities increasing by 2.22%. This indicates that differences between socioeconomic groups became more prominent over time.

Lastly, Madagascar reported a large decrease in within-socioeconomic group inequality for IPTp coverage, with the within-group contribution dropping by 21.104 points. This represents a dramatic improvement in equity within SES groups. Across all countries, the majority of the inequality in ITN use in pregnancy, ITN use by under-fives, and IPTp coverage is driven by within- socioeconomic group disparities rather than between-group differences. The within-group contributions are highest (often at 100%) in most countries, indicating that disparities within people of the same socioeconomic class are the main source of inequality.

## 4. Discussion

The aim of this study was to examine the temporal and socioeconomic inequalities associated with three critical malaria prevention interventions: ITN use during pregnancy, ITN use among children under five, and IPTp coverage across 10 sub-Saharan African countries. Drawing on DHS surveys from two different time points per country, this analysis explored how the distribution of these interventions changed over time and the extent to which socioeconomic status (SES) and place of residence influenced access to these essential malaria intervention tools. Our study findings highlighted considerable variability in the proportions of ITN use during pregnancy, ITN use by children under five, and IPTp coverage both across the 10 countries and over time. While countries like Nigeria and Burkina Faso demonstrated remarkable progress, others, such as Kenya and Rwanda, experienced stagnation or even declines in key indicators, reflecting a nuanced landscape of malaria prevention efforts. This variation aligns with previous research that has documented the uneven success of national malaria programs, largely due to differing levels of healthcare infrastructure, resource allocation, and the effectiveness of implementation strategies [34]. Burkina Faso’s consistent high coverage in both ITN use and IPTp, even in rural areas, underscores the importance of sustained and targeted interventions. Such improvements parallel findings from Bhatt et al. [35], who emphasized that the large-scale distribution of malaria prevention tools, combined with effective public health campaigns, can lead to significant reductions in malaria transmission and improve overall health outcomes in vulnerable populations [35].

Moreover, the trends observed in Kenya and Rwanda, where some indicators either stagnated or declined, suggest that barriers such as unequal healthcare access, geographic disparities, and perhaps lapses in policy implementation could be hindering progress. These results emphasize the need for more context-specific interventions that address the unique challenges faced by different countries and regions within them. This study thus contributes to the growing body of literature indicating that while significant strides have been made in malaria prevention, persistent inequalities, particularly related to socioeconomic status and place of residence, continue to impede universal access to life-saving interventions [36].

The concentration indices (ENCI) for ITN use during pregnancy present a complex picture of socioeconomic inequality in access to malaria prevention interventions. In countries such as Côte d’Ivoire and Nigeria, the results indicate a clear shift toward pro-poor inequality, meaning that ITN use was more concentrated among poorer populations over time. This shift suggests that national malaria programs in these countries have successfully implemented targeted interventions aimed at increasing ITN coverage among disadvantaged groups. Such outcomes align with findings from Talipouo et al. [37], who reported that well-planned distribution campaigns can significantly improve ITN access in low-income communities [37]. Additionally, this observation may also reflect the geographical overlap between poorer populations and rural or less urbanized areas where malaria risk is higher. In these areas, limited infrastructure, poor drainage systems, and favorable environmental conditions for the proliferation of *Anopheles* mosquitoes make malaria a more pressing issue. Conversely, urban areas, which are more likely to have advanced infrastructure, often face lower malaria risk due to the prevalence of *Culex* mosquitoes, which thrive in polluted waters. These geographical and infrastructural dynamics likely complement the success of targeted intervention programs in achieving the observed pro-poor outcomes [38,39,40,41].

However, the results also highlight persistent pro-rich inequality in other countries, notably Rwanda and Kenya, where wealthier households are still more likely to benefit from ITN use during pregnancy. This disparity is concerning because it suggests that despite overall improvements in ITN distribution, the poorest women, who are at the highest risk for malaria-related complications during pregnancy, are not receiving adequate protection. The increase in pro-rich inequality in Kenya over time is particularly troubling, as it implies that gains made in ITN distribution are disproportionately benefiting wealthier populations. This finding echoes the concerns raised by Sumari et al. [42], who emphasized the need for more tailored interventions that specifically target vulnerable populations to close these gaps in access [42]. Without such targeted efforts, the most disadvantaged groups will continue to face barriers to accessing essential malaria prevention tools, which could undermine broader efforts to reduce malaria prevalence and mortality among pregnant women.

In contrast, the concentration indices for IPTp coverage show a more concerning picture of entrenched inequality across most countries studied. The results revealed a consistent pro-rich bias, meaning that wealthier women were more likely to access IPTp services compared with their poorer counterparts. This inequality was particularly pronounced in Cameroon, Mozambique, and Nigeria, where ENCI values indicated increasing inequality over time. The concentration of IPTp services among wealthier women in these countries is concerning, especially given the global emphasis on ensuring universal access to IPTp during pregnancy as part of comprehensive malaria prevention strategies. The results are consistent with previous studies, such as Kalu et al. [43] and Ochieng et al. [44], which found that malaria prevention services, including IPTp, are often more accessible to higher socioeconomic groups, particularly in urban areas where healthcare infrastructure is more developed [43,44].

The persistence of these inequalities in IPTp coverage suggests that systemic barriers, such as inadequate healthcare infrastructure in rural areas, financial constraints, and limited awareness of IPTp benefits, continue to hinder access for poorer women. These findings underscore the need for more inclusive malaria prevention policies that specifically address these barriers. For example, improving access to healthcare services in rural and remote areas, subsidizing IPTp for low-income women, and enhancing community outreach programs could help reduce these disparities [45]. While countries such as Burkina Faso and Madagascar showed slight reductions in IPTp inequality, the overall low coverage of IPTp remains a significant public health challenge. As noted by Chen et al. [46], scaling up IPTp services is essential to reducing the risk of malaria-related complications during pregnancy, particularly in underserved areas where women are most at risk [46]. The findings of this study reinforce the urgency of addressing these inequalities by prioritizing interventions that target low-income and rural populations, where the need for IPTp is greatest.

For ITN use among children under five, the concentration indices revealed stark disparities across the countries studied. In some countries, such as Côte d’Ivoire and Nigeria, there were significant shifts towards pro-poor ITN use, reflecting successful efforts to redistribute ITNs to poorer households. This shift is an encouraging sign that malaria prevention strategies in these countries are beginning to reach the most vulnerable populations, particularly children under five, who are at the greatest risk of severe malaria and death. The pro-poor trend in these countries aligns with findings from earlier studies, such as Maghendji-Nzondo, Nzoughe, Lemamy, Kouna, Pegha-Moukandja, Lekoulou, Mbatchi, Toure-Ndouo, and Lekana-Douki [34], which highlighted the positive impact of targeted ITN interventions on improving access for disadvantaged households [34]. These results suggest that when malaria prevention campaigns are designed to prioritize equity, they can achieve meaningful improvements in ITN coverage among poorer populations, helping to protect children from the worst outcomes of malaria.

However, the analysis also showed that pro-rich inequality persists in countries like Kenya, Mozambique, and Rwanda, where wealthier households continue to have better access to ITNs for their children. This inequality is particularly concerning in countries with high malaria burdens, where children under five are especially vulnerable to severe malaria and related mortality. The continued concentration of ITN use among wealthier households in these countries suggests that malaria prevention campaigns have not been equally effective across all socioeconomic groups. This may be due to several factors, including geographic disparities in healthcare access, differences in awareness and education about ITNs [47], and financial barriers that prevent poorer families from acquiring sufficient ITNs. These findings are consistent with previous research, such as Talipouo, Ngadjeu, Doumbe-Belisse, Djamouko-Djonkam, Sonhafouo-Chiana, Kopya, Bamou, Awono-Ambene, Woromogo, and Kekeunou [37], which emphasized the importance of prioritizing ITN distribution in the poorest and most rural communities to achieve more equitable health outcomes [37].

Theil indices for ITN use in pregnancy, ITN use by children under five, and IPTp coverage revealed critical insights into the unequal distribution of malaria prevention services based on place of residence. In countries such as Mozambique, Côte d’Ivoire, and Nigeria, the analysis showed higher levels of inequalities in rural populations compared with their urban counterparts, particularly concerning IPTp coverage. This disparity in access to malaria prevention services between rural and urban areas has been well-documented in earlier studies, such as Assele et al. [48], which identified rural residence as a significant determinant of limited access to essential health services, including malaria prevention tools [48]. The higher inequality in rural areas highlights the challenges faced by these communities, such as poor healthcare infrastructure, geographical barriers, and limited access to information on malaria prevention interventions [24].

However, a deeper examination through the decomposition of the Theil indices revealed that within-group inequality, rather than between-group inequality, was the predominant driver of overall disparities in most countries. This meant that while differences between rural and urban areas exist, the more significant issue lay within these groups, where the distribution of malaria prevention interventions is highly uneven. For instance, within both rural and urban populations, there are subgroups perhaps distinguished by income, education, or other socioeconomic factors that experienced vastly different levels of access to ITN use and IPTp coverage. This within-group inequality is a critical finding, as it suggests that targeting broad rural or urban populations without addressing these internal disparities may not be sufficient to reduce overall inequality. These results are in line with findings from Imboumy-Limoukou et al. [49], who also emphasized the importance of addressing intra-group disparities to achieve more equitable health outcomes [49].

Theil indices by socioeconomic status (SES) further underscored the disparities in ITN use and IPTp coverage across different income groups. In countries like Nigeria, Burkina Faso, and Côte d’Ivoire, the data show significant improvements, with overall inequality decreasing over time. These countries have made strides in reducing disparities in access to malaria prevention tools, likely due to targeted programs that have successfully reached the poorest households. In contrast, countries like Kenya and Rwanda have seen an increase in inequality, particularly regarding IPTp coverage. This trend is concerning, as it indicates that efforts to scale up malaria prevention interventions in these countries may not be reaching the most disadvantaged populations. Such findings are consistent with the observations of Maghendji-Nzondo, Nzoughe, Lemamy, Kouna, Pegha-Moukandja, Lekoulou, Mbatchi, Toure-Ndouo, and Lekana-Douki [34], who documented similar patterns of increasing inequality in access to healthcare services in these countries [34]. The growing inequality in IPTp coverage, especially in Kenya and Rwanda, points to systemic barriers that disproportionately affect poorer women, such as financial constraints, inadequate healthcare infrastructure in rural areas, and lack of awareness about the benefits of IPTp [2,50].

Importantly, the persistence of within-group inequality among lower socioeconomic groups suggests that even within the poorest populations, there are subgroups that face more significant barriers to accessing malaria prevention interventions. These within-group disparities indicate that not all poor households are equally disadvantaged as some may still have relatively better access to services than others, due to factors like proximity to healthcare facilities, education, or community support networks. These finding highlighted the need for more targeted and finely tuned interventions that address the unique challenges faced by different subgroups within the poorest populations. As argued by multiple authors, simply categorizing populations by socioeconomic status without recognizing the heterogeneity within these groups may obscure the true nature of health disparities and hinder the effectiveness of public health interventions [51,52,53]. Policies aimed at reducing inequality in malaria prevention must therefore consider the diverse needs of subpopulations within broader socioeconomic categories, ensuring that the most marginalized individuals receive adequate attention.

The decomposition of Theil indices by both place of residence and SES provides further insights into the complex nature of inequality in malaria prevention. Across most countries, between-group inequality, whether between rural and urban areas or between different SES groups, contributed very little to the overall inequality. Instead, within-group inequalities remained the primary driver of disparities in ITN use and IPTp coverage. These findings are crucial, as they suggest that while geographic and socioeconomic differences are important, the most significant inequalities are found within these larger population categories. This trend echoes the conclusions of Imboumy-Limoukou, Maghendji-Nzondo, Sir-Ondo-Enguier, Niemczura De Carvalho, Tsafack-Tegomo, Buekens, Okouga, Mouinga-Ondeme, Kwedy Nolna, and Lekana-Douki [49], who emphasized that addressing disparities within population subgroups is essential for improving equity in access to malaria prevention services [49]. Focusing solely on broad categories like “rural” or “poor” may miss the difference and deeply ingrained inequalities that exist within these groups, which can perpetuate health disparities despite efforts to increase overall coverage.

However, for IPTp coverage specifically, the increase in between-group inequality observed in countries like Cameroon and Mozambique highlights a troubling trend of growing disparities between socioeconomic classes. This suggests that while overall IPTp coverage may be improving, wealthier women are benefiting disproportionately, exacerbating inequalities between the rich and poor. Addressing this issue will require more than just expanding IPTp services; it will necessitate a concerted effort to reduce socioeconomic inequalities by ensuring that the poorest and most vulnerable women have equal access to these life-saving interventions. In rural and underserved areas, where malaria prevalence is often highest, weak infrastructure can hinder the distribution and sustained use of these interventions, thus worsening inequalities [54,55]. This highlights the need to bolster health infrastructure in such settings including but not limited to well-equipped health centers, supply chains, and trained healthcare personnel, to enhance service delivery and builds trust within communities. These findings are a clear reminder that achieving equity in malaria prevention requires not only increasing coverage but also ensuring that the benefits of these interventions are distributed fairly across all population groups. Without addressing both within-group and between-group inequalities, the goal of universal access to malaria prevention tools will remain elusive [56].

To address these persistent disparities, it is essential for malaria prevention efforts to focus on expanding ITN access in underserved areas, particularly rural and low-income communities where the need is greatest [57]. This could involve strengthening community-based distribution programs, improving healthcare infrastructure in remote areas, and increasing education and awareness campaigns that target vulnerable populations. Without concerted efforts to close these gaps, the benefits of ITN distribution will remain unevenly distributed, and the most disadvantaged children will continue to face an elevated risk of malaria-related illness and death [58]. Overall, the findings from the concentration indices of ITN use in pregnancy, IPTp coverage, and ITN use among children under five highlight the critical importance of addressing socioeconomic inequalities in malaria prevention efforts. While some countries have made notable progress in reducing these disparities, others continue to struggle with pro-rich inequality, which limits the effectiveness of malaria prevention strategies. Addressing these inequalities will require a coordinated effort to design and implement more inclusive policies that prioritize the needs of the poorest and most vulnerable populations. By doing so, countries can improve access to life-saving malaria prevention tools and reduce the burden of malaria on pregnant women and young children across sub-Saharan Africa.

### Limitations and Strengths

This study is not without limitations. First, this study utilized data from cross-sectional surveys, hence it is impossible to draw causal links between ITN use and IPTp coverage and SES/residence status. Second, while this study offered the opportunity to situate temporal trends of the outcomes in two time periods, the same cannot be said of individual level changes. Third, the Theil index, while offering policy-relevant insights into between- and within-group inequalities, is more sensitive to changes at the extremes of socioeconomic distribution compared with other indices like the Gini coefficient, and as such, it may overstate inequalities in cases where extremes are not representative of the larger population. Additionally, the Theil index has a lower bound of zero and this may mask nuances in relatively equal distributions as small variations in inequality are harder to detect compared with other measures [59]. Finally, cross-country comparisons were not feasible due to variations in the base and end years across countries.

Despite these limitations, this study has multiple strengths. First, because of utilizing high quality, nationally representative, and the most-recent cross-country DHS datasets, the findings from this study are robust and can be generalized across multiple contexts in SSA. Second, this study utilized both the ENCI and Theil indices in examining inequalities. This comprehensive approach offers multidimensional inequality insights. When combined, ENCI and the Theil index allow researchers to simultaneously assess inequalities in health outcomes (via the ENCI) and wealth distribution inequalities (via the Theil index), providing insights into whether health inequalities are driven primarily by economic factors or other social determinants [32,33,60]. This allows for a more nuanced understanding of inequalities which is critical in policy decisions.

## 5. Conclusions

This study has provided valuable insights into the temporal inequalities in the uptake of malaria prevention strategies among pregnant women and children under five across 10 sub-Saharan African countries. The findings highlight significant within-country variations with some countries demonstrating notable progress, while others experienced widening disparities. Generally, pregnant women and children in poor households and those in rural areas continue to face barriers in accessing and utilizing preventive interventions. Temporal analysis reveals that although inequality may have reduced in some areas, in other contexts, like Kenya and Rwanda, inequalities are widening which demands urgent targeted and sustained policy efforts if these countries are looking to make progress towards the Sustainable Development Goals.

Addressing within-country disparities is crucial for achieving universal health coverage and reducing malaria-related morbidity and mortality in the region. Policymakers must prioritize interventions that not only increase the overall coverage of malaria prevention strategies but also ensure that they identify and reach the most vulnerable populations living in both rural and urban areas and across the various SES groups. Lastly, the continuous monitoring and evaluation of inequality trends over time, coupled with inclusive health policies, will be essential to ensure that no woman or child is left behind in the fight against malaria in sub-Saharan Africa.

## Figures and Tables

**Figure 1 ijerph-21-01656-f001:**
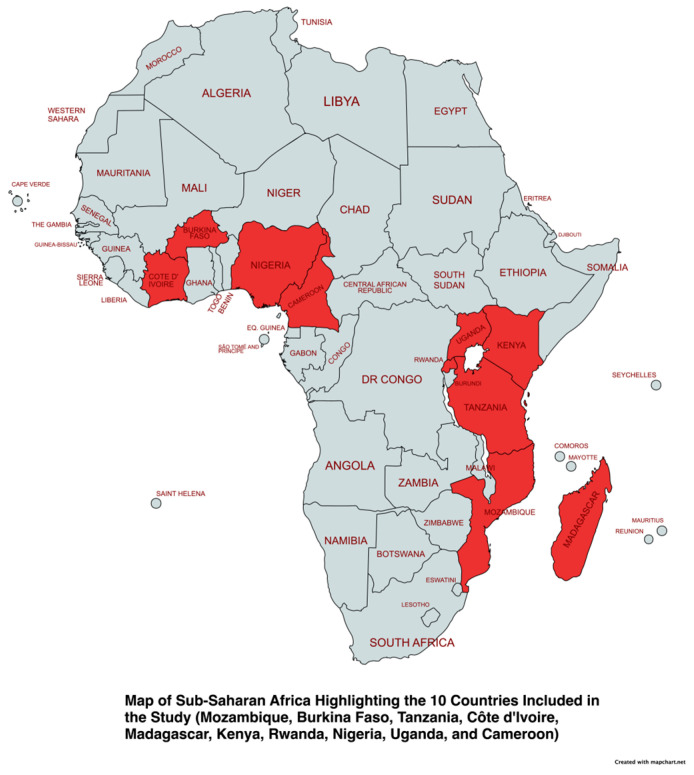
Map of Sub-Saharan Africa Highlighting the 10 Countries Included in this Study.

**Table 1 ijerph-21-01656-t001:** A summary of between-group and within-group inequality.

Dimension	Between-Group Inequality	Within-Group Inequality
Scope	Across distinct groups (e.g., Q1–Q5 or Rural versus Urban living)	Among individuals within the same group (e.g., people living in rural areas)
Analysis objective	Variations due to differences in group means	Variations among individuals sharing characteristics
Key insight	Explains inequality attributable to group membership	Highlights heterogeneity within groups
Policy focus	Group-targeted interventions	Individualized or subgroup-focused interventions

**Table 2 ijerph-21-01656-t002:** The proportions of ITN use in pregnancy, ITN use by children under-5, and IPTp coverage for 10 countries in SSA, various years.

Country	Year	ITN Use in Pregnancy % [95% CI]	% Change in ITN Use in Pregnancy	ITN Use by Under-Fives % [95% CI]	% Change in ITN Use in Under-Fives	IPTp Coverage % [95% CI]	% Change in IPTp Coverage
Mozambique	2011	61.07 [56.60;65.53]	+18.80	65.15 [62.77;67.53]	+19.63	9.63 [8.16;11.11]	+159.40
	2022/3	72.55 [67.70;77.39]		77.94 [75.42;80.45]		24.98 [22.75;27.21]	
Burkina Faso	2010	76.65 [73.40;79.89]	+8.39	76.99 [75.20;78.77]	+13.46	40.95 [38.79;43.10]	+38.41
	2021	83.08 [80.37;85.79]		87.35 [86.16;88.54]		56.68 [54.21;59.14]	
Tanzania	2015/16	74.40 [70.69;78.11]	+13.74	76.71 [74.67;78.75]	+13.86	7.87 [6.71;9.03]	+299.87
	2022	84.62 [81.24;88.00]		87.34 [86.20;88.49]		31.47 [29.13;33.81]	
Côte d’Ivoire	2011/12	57.11 [52.11;62.12]	+36.89	59.61 [56.79;62.41]	+29.56	8.16 [6.64;9.67]	+322.18
	2021	78.18 [74.49;81.88]		77.23 [74.72;79.73]		34.45 [31.70;37.20]	
Madagascar	2008/09	75.86 [71.94;79.78]	+1.78	80.38 [78.46;82.31]	+5.54	1.93 [1.36;2.49]	+1504.15
	2021	77.21 [73.81;80.62]		84.83 [83.20;86.46]		30.96 [28.80;33.11]	
Kenya	2014	76.30 [72.93;79.67]	−1.85	80.97 [79.72;82.20]	−1.52	10.55 [9.25;11.85]	−38.86
	2022	74.89 [71.29;78.49]		79.74 [78.36;81.12]		6.45 [5.68;7.22]	
Nigeria	2013	29.82 [26.82;32.81]	+173.88	31.48 [29.16;33.80]	+153.97	7.02 [6.17;7.88]	+136.75
	2018	81.67 [79.67;83.68]		79.95 [78.70;81.19]		16.62 [15.51;17.72]	
Uganda	2011	71.14 [66.79;75.48]	+11.12	71.58 [69.42;73.74]	+14.64	10.57 [9.07;12.07]	+57.24
	2016	79.05 [76.53;81.57]		82.06 [80.79;83.32]		17.40 [16.16;18.63]	
Cameroon	2011	46.44 [39.86;53.01]	+67.44	51.17 [47.80;83.32]	+56.65	12.13 [10.90;13.36]	+168.26
	2018	77.76 [74.04;81.48]		80.16 [78.22;82.10]		32.54 [30.26;34.83]	
Rwanda	2014/15	87.72 [85.31;90.12]	−6.76	86.69 [85.48;87.90]	−3.20	-	-
	2019/20	81.79 [78.13;85.46]		83.92 [82.42;85.43]		-	

Note: The interpretations provided focus on within-country trends over time and avoid cross-country comparison, as such may be misleading.

**Table 3 ijerph-21-01656-t003:** Erreygers Normalized Concentration indices (ENCI) for ITN use in pregnancy for 10 countries in SSA, various years.

Country	Year	ENCI	*p*-Values	Robust Standard Errors
Mozambique	2011	−0.003	0.934	0.040
	2022/3	0.000	0.995	0.040
Burkina Faso	2010	−0.003	0.914	0.029
	2021	−0.023	0.334	0.024
Tanzania	2015/16	−0.006	0.862	0.035
	2022	−0.008	0.761	0.027
Côte d’Ivoire	2011/12	−0.117	0.003	0.040
	2021	−0.216	0.000	0.028
Madagascar	2008/09	−0.002	0.950	0.031
	2021	−0.049	0.108	0.030
Kenya	2014	0.069	0.009	0.026
	2022	0.052	0.063	0.028
Rwanda	2014/15	0.109	0.000	0.026
	2019/20	0.122	0.001	0.036
Nigeria	2013	0.067	0.010	0.026
	2018	−0.156	0.000	0.018
Uganda	2011	−0.037	0.330	0.038
	2016	0.025	0.248	0.022
Cameroon	2011	0.090	0.148	0.061
	2018	0.020	0.543	0.034

Note: The interpretations provided focus on within-country trends over time and avoid cross-country comparison, as such may be misleading.

**Table 4 ijerph-21-01656-t004:** Erreygers Normalized Concentration indices (ENCI) for IPTp coverage for 10 countries in SSA, various years.

Country	Year	ENCI	*p*-Values	Robust Standard Errors
Mozambique	2011	0.043	0.001	0.011
	2022/3	0.149	0.000	0.018
Burkina Faso	2010	0.084	0.000	0.017
	2021	0.082	0.000	0.021
Tanzania	2015/16	0.016	0.113	0.010
	2022	0.033	0.085	0.019
Côte d’Ivoire	2011/12	0.035	0.004	0.012
	2021	0.117	0.000	0.019
Madagascar	2008/09	−0.003	0.611	0.005
	2021	0.062	0.001	0.019
Kenya	2014	0.016	0.080	0.009
	2022	0.019	0.005	0.008
Nigeria	2013	0.026	0.000	0.006
	2018	0.115	0.000	0.010
Uganda	2011	0.012	0.329	0.012
	2016	0.008	0.454	0.011
Cameroon	2011	0.080	0.000	0.012
	2018	0.178	0.000	0.019

Note: The interpretations provided focus on within-country trends over time and avoid cross-country comparison, as such may be misleading.

**Table 5 ijerph-21-01656-t005:** Erreygers Normalized Concentration indices (ENCI) for ITN use for 10 countries in SSA, various years.

Country	Year	ENCI	*p*-Values	Robust Standard Errors
Mozambique	2011	−0.059	0.006	0.022
	2022/3	0.054	0.006	0.020
Burkina Faso	2010	0.003	0.843	0.014
	2021	−0.022	0.020	0.010
Tanzania	2015/16	0.052	0.003	0.018
	2022	0.009	0.381	0.011
Côte d’Ivoire	2011/12	−0.090	0.000	0.021
	2021	−0.183	0.000	0.014
Madagascar	2008/09	0.020	0.148	0.014
	2021	−0.005	0.642	0.012
Kenya	2014	0.086	0.000	0.010
	2022	0.027	0.006	0.010
Rwanda	2014/15	0.068	0.000	0.011
	2019/20	0.080	0.000	0.013
Nigeria	2013	0.060	0.002	0.019
	2018	−0.139	0.000	0.011
Uganda	2011	0.010	0.535	0.016
	2016	0.010	0.300	0.010
Cameroon	2011	0.083	0.014	0.034
	2018	−0.044	0.009	0.017

Note: The interpretations provided focus on within-country trends over time and avoid cross-country comparison, as such may be misleading.

**Table 6 ijerph-21-01656-t006:** Theil indices for subgroups for ITN use and IPTp coverage distinguished by place of residence for 10 countries in SSA, various years.

Country	Year	ITN Use in Pregnancy		IPTp Coverage		ITN Use by Under-Fives	
		Urban GE (2)	Rural GE (2)	Urban GE (2)	Rural GE (2)	Urban GE (2)	Rural GE (2)
Mozambique	2011	0.215	0.444	0.326	0.674	0.236	0.345
	2022/3	0.164	0.221	1.005	1.874	0.099	0.189
Burkina Faso	2010	0.171	0.138	0.552	0.694	0.159	0.134
	2021	0.120	0.090	0.331	0.390	0.088	0.067
Tanzania	2015/16	0.145	0.164	4.386	7.084	0.130	0.162
	2022	0.094	0.103	1.077	1.639	0.077	0.080
Côte d’Ivoire	2011/12	0.429	0.295	5.161	7.592	0.407	0.285
	2021	0.172	0.096	0.940	1.207	0.186	0.090
Madagascar	2008/09	0.136	0.166	22.53	22.26	0.093	0.131
	2021	0.187	0.127	1.083	1.008	0.086	0.086
Kenya	2014	0.113	0.186	4.080	4.300	0.095	0.141
	2022	0.151	0.159	6.467	7.657	0.139	0.118
Nigeria	2013	0.891	1.273	6.905	7.281	0.834	1.145
	2018	0.179	0.093	1.880	2.848	0.179	0.106
Uganda	2011	0.097	0.209	5.272	4.777	0.148	0.205
	2016	0.109	0.137	2.689	2.363	0.089	0.113
Cameroon	2011	0.426	0.645	2.579	4.279	0.383	0.488
	2018	0.159	0.178	0.733	1.443	0.138	0.126
Rwanda	2014/15	0.046	0.079	-	-	0.045	0.086
	2019/20	0.058	1.275	-	-	0.066	0.107

Note: Data for IPTp coverage was not available for Rwanda. Note: The interpretations provided focus on within-country trends over time and avoid cross-country comparison, as such may be misleading.

**Table 7 ijerph-21-01656-t007:** Theil indices for subgroups for ITN use and IPTp coverage distinguished by SES for 10 countries in SSA, various years.

Country	Year	ITN Use in Pregnancy			IPTp Coverage			ITN Use by Under-Fives		
		Poor GE (2)	Middle GE (2)	Rich GE (2)	Poor GE (2)	Middle GE (2)	Rich GE (2)	Poor GE (2)	Middle GE (2)	Rich GE (2)
Mozambique	2011	0.374	0.277	0.374	6.964	5.116	3.970	0.269	0.257	0.345
	2022/3	0.190	0.226	0.194	2.315	1.714	0.994	0.179	0.162	0.130
Burkina Faso	2010	0.141	0.153	0.145	0.791	0.654	0.551	0.146	0.132	0.140
	2021	0.094	0.086	0.110	0.445	0.365	0.313	0.065	0.068	0.083
Tanzania	2015/16	0.167	0.120	0.173	6.688	7.531	5.287	0.142	0.162	0.158
	2022	0.104	0.078	0.109	1.584	1.458	1.320	0.074	0.080	0.083
Côte d’Ivoire	2011/12	0.268	0.369	0.450	7.923	9.047	4.405	0.280	0.283	0.432
	2021	0.063	0.153	0.285	1.335	1.187	0.684	0.071	0.147	0.258
Madagascar	2008/09	0.155	0.203	0.149	20.80	24.32	23.30	0.127	0.120	0.117
	2021	0.119	0.137	0.161	1.201	0.815	0.922	0.084	0.089	0.087
Kenya	2014	0.184	0.164	0.120	4.664	3.289	4.114	0.153	0.112	0.095
	2022	0.186	0.106	0.146	8.606	5.975	6.331	0.120	0.094	0.150
Nigeria	2013	1.403	0.852	1.047	10.06	4.841	6.422	1.244	0.800	0.961
	2018	0.072	0.121	0.226	3.649	2.465	1.541	0.085	0.125	0.215
Uganda	2011	0.163	0.204	0.197	5.365	4.259	4.671	0.173	0.245	0.186
	2016	0.130	0.185	0.104	2.482	2.394	2.338	0.104	0.126	0.103
Cameroon	2011	0.650	0.542	0.445	4.851	3.796	2.281	0.527	0.426	0.375
	2018	0.171	0.199	0.150	1.566	1.120	0.649	0.113	0.146	0.143
Rwanda	2014/15	0.127	0.044	0.045	-	-	-	0.100	0.081	0.052
	2019/20	0.184	0.094	0.076	-	-	-	0.128	0.087	0.078

**Table 8 ijerph-21-01656-t008:** Decomposition of Theil indices by place of residence.

Country	Year	ITN Use in Pregnancy			IPTp Coverage			ITN Use by Under-Fives		
		Overall Inequality GE (2)	${GE}_{B}(2)$ (%Contr)	${GE}_{w}\left(2\right)$ (%Contr)	Overall Inequality GE (2)	${GE}_{B}(2)$ (%Contr)	${GE}_{w}\left(2\right)$ (%Contr)	Overall Inequality GE (2)	${GE}_{B}(2)$ (%Contr)	${GE}_{w}\left(2\right)$ (%Contr)
Mozambique	2011	0.353	0.009 (2.55%)	0.344 (97.45%)	5.01	0.021 (0.42%)	4.989 (99.58%)	0.301	0.002 (0.33%)	0.300 (99.67%)
	2022/3	0.199	0.000 (0%)	0.199 (100%)	1.517	0.026 (1.71%)	1.491 (98.29)	0.151	0.002 (1.32%)	0.149 (98.68%)
Burkina Faso	2010	0.145	0.000 (0%)	0.145 (100%)	0.659	0.001 (0.15%)	0.658 (99.85)	0.140	0.000 (0%)	0.140 (100%)
	2021	0.099	0.000 (0%)	0.099 (100%)	0.372	0.001 (0%)	0.372 (100%)	0.073	0.000 (0%)	0.073 (100%)
Tanzania	2015/16	0.159	0.000 (0%)	0.159 (100%)	6.202	0.021 (0.34%)	6.181 (99.66%)	0.153	0.000 (0%)	0.153 (100%)
	2022	0.101	0.000 (0%)	0.101 (100%)	1.449	0.010 (0.69%)	1.439 (99.31%)	0.080	0.000 (0%)	0.080 (100%)
Côte d’Ivoire	2011/12	0.334	0.003 (0.60%)	0.332 (99.40%)	6.555	0.016 (0.24%)	6.539 (99.76%)	0.321	0.002 (0.62%)	0.319 (99.38%)
	2021	0.126	0.002 (1.59%)	0.124 (98.41%)	1.088	0.004 (0.37%)	1.084 (99.63%)	0.123	0.003 (1.63%)	0.121 (98.37%)
Madagascar	2008/09	0.160	0.000 (0%)	0.160 (100%)	22.12	0.000 (0%)	22.12 (100%)	0.122	0.000 (0%)	0.122 (100%)
	2021	0.136	0.001 (0%)	0.136 (100%)	1.021	0.000 (0%)	1.021 (100%)	0.086	0.000 (0%)	0.086 (100%)
Kenya	2014	0.158	0.002 (0.67%)	0.157 (99.37%)	4.227	0.000 (0%)	4.227 (100%)	0.124	0.001 (0.81%)	0.123 (99.19%)
	2022	0.156	0.000 (0%)	0.156 (100%)	7.214	0.003 (0%)	7.212 (100%)	0.125	0.000 (0%)	0.125 (100%)
Nigeria	2013	0.148	0.007 (4.73%)	1.141 (95.27%)	7.152	0.000 (0%)	7.152 (100%)	1.036	0.005 (0.48%)	1.031 (99.52%)
	2018	0.116	0.002 (1.72%)	0.114 (98.28%)	2.412	0.016 (0.66%)	2.396 (99.34%)	0.128	0.001 (0.78%)	0.127 (99.22%)
Uganda	2011	0.182	0.003 (1.65%)	0.179 (98.35%)	4.879	0.001 (0.02%)	4.878 (99.98)	0.189	0.000 (0%)	0.189 (100%)
	2016	0.131	0.000 (0%)	0.131 (100%)	2.417	0.000 (0%)	2.417 (100%)	0.108	0.000 (0%)	0.108 (100%)
Cameroon	2011	0.544	0.006 (1.10%)	0.538 (98.90%)	3.411	0.025 (0.73%)	3.386 (99.27%)	0.440	0.002 (0.45%)	0.438 (99.55%)
	2018	0.169	0.000 (0%)	0.169 (100%)	1.051	0.026 (2.47%)	1.025 (97.53%)	0.132	0.000 (0%)	0.132 (100%)
Rwanda	2014/15	0.071	0.000 (0%)	0.071 (100%)	-	-	-	0.075	0.000 (0%)	0.075 (100%)
	2019/20	0.109	0.001 (0.92%)	0.108 (99.08%)	-	-	-	0.096	0.000 (0%)	0.096 (100%)

Note: Data for IPTp coverage were not available for Rwanda. Note: The interpretations provided focus on within-country trends over time and avoid cross-country comparison, as such may be misleading.

**Table 9 ijerph-21-01656-t009:** Decomposition of Theil indices by SES.

Country	Year	ITN Use in Pregnancy			IPTp Coverage			ITN Use by Under-Fives		
		Overall Inequality GE (2)	${GE}_{B}(2)$ (%Contr)	${GE}_{w}\left(2\right)$ (%Contr)	Overall Inequality GE (2)	${GE}_{B}(2)$ (%Contr)	${GE}_{w}\left(2\right)$ (%Contr)	Overall Inequality GE (2)	${GE}_{B}(2)$ (%Contr)	${GE}_{w}\left(2\right)$
Mozambique	2011	0.353	0.001 (0.28%)	0.352 (99.72%)	5.010	0.024 (0.48%)	4.986 (99.52%)	0.300	0.001 (0%)	0.300 (100%)
	2022/3	0.200	0.000 (0%)	0.200 (100%)	1.517	0.041 (2.70%)	1.476 (97.30%)	0.151	0.000 (0%)	0.151 (100%)
Burkina Faso	2010	0.145	0.000 (0%)	0.145 (100%)	0.659	0.004 (0.61%)	0.655 (99.39%)	0.140	0.000 (0%)	0.140 (100%)
	2021	0.099	0.000 (0%)	0.099 (100%)	0.372	0.002 (0.54%)	0.370 (99.46%)	0.073	0.000 (0%)	0.073 (100%)
Tanzania	2015/16	0.159	0.001 (0.63%)	0.158 (99.37%)	6.202	0.008 (0.13%)	6.194 (99.87%)	0.152	0.000 (0%)	0.152 (100%)
	2022	0.100	0.000 (0%)	0.100 (100%)	1.449	0.002 (0.07%)	1.448 (99.93%)	0.079	0.000 (0%)	0.079 (100%)
Côte d’Ivoire	2011/12	0.334	0.004 (1.20%)	0.330 (98.80%)	6.555	0.042 (0.64%)	6.513 (99.36%)	0.321	0.003 (0.93%)	0.318 (99.07%)
	2021	0.126	0.008 (7.14%)	0.117 (92.86%)	1.088	0.019 (1.84%)	1.068 (98.16%)	0.123	0.006 (4.88%)	0.117 (95.12%)
Madagascar	2008/09	0.160	0.000 (0%)	0.160 (100%)	22.13	0.002 (0.05%)	22.12 (99.95%)	0.122	0.000 (0%)	0.122 (100%)
	2021	0.136	0.000 (0%)	0.136 (100%)	1.022	0.006 (0.59%)	1.016 (99.41%)	0.086	0.000 (0%)	0.086 (100%)
Kenya	2014	0.158	0.001 (5.07%)	0.157 (94.93%)	4.227	0.007 (0.17%)	4.220 (99.83%)	0.124	0.001 (0.81%)	0.123 (99.19%)
	2022	0.156	0.001 (0%)	0.156 (100%)	7.214	0.011 (0.17%)	7.202 (99.83%)	0.124	0.000 (0%)	0.124 (100%)
Nigeria	2013	1.147	0.010 (0.87%)	1.137 (99.13%)	7.152	0.038 (0.52%)	7.115 (99.48%)	1.036	0.007 (0.68%)	1.029 (99.32%)
	2018	0.116	0.004 (4.31%)	0.111 (95.69%)	2.412	0.050 (2.07%)	2.362 (97.93%)	0.128	0.004 (2.34%)	0.125 (97.66%)
Uganda	2011	0.181	0.000 (0%)	0.181 (100%)	4.879	0.003 (0.06%)	4.876 (99.94%)	0.190	0.001 (0%)	0.190 (100%)
	2016	0.130	0.000 (0%)	0.130 (100%)	2.417	0.000 (0%)	2.417 (100%)	0.108	0.000 (0%)	0.108 (100%)
Cameroon	2011	0.544	0.004 (0.74%)	0.540 (99.26%)	3.411	0.045 (2.32%)	3.366 (98.68%)	0.440	0.002 (0.68%)	0.437 (99.32%)
	2018	0.169	0.000 (0%)	0.169 (100%)	1.051	0.034 (3.24%)	1.017 (96.76%)	0.131	0.000 (0%)	0.131 (100%)
Rwanda	2014/15	0.071	0.002 (2.82%)	0.069 (97.18%)	_	_	_	0.076	0.001 (1.32%)	0.075 (98.68%)
	2019/20	0.110	0.003 (2.73%)	0.107 (97.27%)	_	_	_	0.096	0.001 (1.04%)	0.095 (98.96%)

Note: The interpretations provided focus on within-country trends over time and avoid cross-country comparison, as such may be misleading.

## Data Availability

Datasets are publicly available at https://dhsprogram.com/data/available-datasets.cfm (accessed on 10 July 2024).

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
