# Peer review of "Malaria Prevention for Pregnant Women and Under-Five Children in 10 Sub-Saharan Africa Countries: Socioeconomic and Temporal Inequality Analysis"

_ijerph, 2024, doi:10.3390/ijerph21121656_

Round 1
Reviewer 1 Report
Comments and Suggestions for Authors
General comments
The article is well written for the most part, and follows a progress ITN and IPTp coverage in a number of SSA countries. However, it can be improved based on the recommendations below.
The authors need to discuss the rationale behind comparing these two distinct groups - pregnant women and children under five - other than their shared vulnerability to malaria. These groups differ significantly in their level of exposure, responsibility for ITN use, and the nature of malaria prevention campaigns targeted to them. For young children, ITN use is contingent upon family adherence to malaria prevention practices, while for pregnant women, it is largely a matter of personal adherence. Their risks and access to malaria prevention resources also differ according to cultural, social, and economic factors. Without proper context or justification, directly comparing these two groups is similar to comparing apples to tomatoes and may not provide a meaningful measure of program effectiveness.
There is also inconsistency in the definitions of "pregnant women" used for IPTp and ITN comparisons. For IPTp coverage, pregnant women are defined as those who have given birth in the past two years, while for ITN use, the definition is expanded to include all women aged 15-49, regardless of pregnancy status. This inconsistency introduces considerable bias and diminishes the validity of the comparisons. The authors should limit their definition of "pregnant women" to women who reported being pregnant at the time of the study, which provides relevant variables, and can then be comparable between IPTp and ITN coverages. Otherwise the results are misleading.
The terms ‘between groups’ and ‘within groups’ are not clearly defined in the paper, which limits one’s ability to interpret the findings. ‘Between groups’ could refer to distinctions across socioeconomic categories, geographic locations, or demographic categories (eg pregnant women vs under fives groups), or even between countries. Similarly, ‘within groups’ could refer to any variations among individuals within any of these categories. The authors should specify which groups are included in ‘between’ and ‘within’ comparisons. This information can be tabulated in the introductions section. Similarly, please also define the terms ‘equality’ and ‘inequality’ in comparisons; these definitions can also be provided in the intro section.
Specific comments
Line 48: What does violence malaria mean? I have never before heard that term
Lines 119 – 123: Please provide more details on the inequalities; what groups were more favored than others? Were the comparisons between children and pregnant women? Was there a consideration for the different modes of distribution?
Line 149: What do you mean when saying between groups? If you are focusing on pregnant women, what other groups are you comparing with? Or are you comparing between under fives and pregnant women? Do you have any way of accounting for the different modes of ITN distributions, risk of exposure or incentives for use between these groups?
Lines 187 – 188: Is there something missing here? These are women of child-bearing age, not necessarily pregnant. Did you assess use by actual pregnant women?
Lina 288: Definition of ITN use in pregnancy is unclear, as definition provided above refers to ITN use by women of child-bearing age, not necessarily pregnant. This can be quite misleading.
Lines 288 - 300: These are all very interesting findings. However, for a fair comparison between the countries, compare within the same years. For example if you are using the baseline data of 2013, let that be uniform across the countries (which I believe it is), and for the current year, make that uniform across the countries, which is not the case here. For some countries you are showing results from 2018, and for others 2021/2022. Make that uniform across the countries.
Lines 303 – 310: Again, there could be a bias here, as you are comparing IPTp use among pregnant women and ITN use among women of child-bearing age.
Lines 318 – 319: Again, consider uniformity in reporting the cases; If it is between 2013 and 2018, or between 2013 and 2022, it should be for all countries, otherwise it cannot be comparable
Tables 1 - 8: The tables are quite confusing; you cannot compare between countries using data from different years. As expressed above, this approach introduces bias, as it fails to account for temporal differences in intervention rollouts, economic conditions, and other influential factors that vary year by year. These discrepancies make the cross-country comparisons less meaningful and risk drawing inaccurate conclusions about each country’s progress relative to others. For a more meaningful comparison, please use data from the same time period across all countries. This comment applies to the figures as well. If you lack data for some time-periods for respective countries, it is best that you take those out of the comparison, but oyu cannot compare different years for the reasons provided above.
Line 413 – 425: Again, these comparisons are biased when you are not complying to the same time-periods. This comment applies to a lot of the paragraphs where country comparisons have been made
Line 430: Again, what does between group means really? Pregnant vs Children? Or pregnant between countries? Or pregnant ITN vs pregnant IPTp? Please provide more clarity
Lines 456 – 492: Also, provide more explanation of equality or inequality actually mean? Again, what does between and within groups mean?
Reviewer 2 Report
Comments and Suggestions for Authors
General comment:
This is a well-written and highly engaging paper, and I believe it will make a significant contribution to a better understanding of socioeconomic barriers and inequalities, as well as disparities between advantaged and disadvantaged settings, in the use of insecticide-treated nets and IPTp. However, I have a few minor comments, which are outlined below.
Major Comment: NA
Minors comments:
Comment 1:
Regarding the paragraph "However, for IPTp coverage specifically, the increase in between-group inequality 684 observed in countries like Cameroon and Mozambique highlights a troubling trend of 685 growing disparities between socioeconomic classes. ….than just expanding IPTp services; it will necessitate a concerted effort to reduce socioeconomic inequalities by ensuring that the poorest and most vulnerable women have equal 690 access to these life-saving interventions.
Could the observed disparities in ITNcoverage between socioeconomic groups also be attributed to the uneven distribution of infrastructure improvements in rural and remote areas? Efforts to reduce socioeconomic inequalities in access to ITN might also require targeted investments in improving healthcare infrastructure in these underserved areas. Given that malaria is more prevalent in such regions, should the focus not be on prioritizing these areas in the fight against malaria to ensure equitable access to life-saving interventions?
Comment 2
Concerning the following paragraph "The concentration indices (ENCI) for ITN use during pregnancy present a complex picture of socioeconomic inequality in access to malaria prevention interventions. In countries such as Côte d'Ivoire and Nigeria, the results indicate a clear shift toward pro-poor inequality, meaning that ITN use was more concentrated among poorer populations over time. This shift suggests that national malaria programs in these countries have success fully implemented targeted interventions aimed at increasing ITN coverage among disadvantaged groups.
Could this observation also be related to the nature of the areas where insecticide-treated nets (ITNs) are predominantly used? Malaria is more closely associated with low-infrastructure, often rural or non-urbanized settings, where the absence of drainage systems and other basic infrastructure creates favorable conditions for the proliferation of Anopheles mosquitoes, the primary malaria vectors. In contrast, urban areas are more likely to experience a prevalence of Culex mosquitoes, which thrive in polluted waters and environments with advanced road and urban infrastructure. Could the concentration of ITN use among poorer populations also reflect a geographical overlap with these less urbanized areas, rather than being solely an outcome of targeted intervention programs?
Reviewer 3 Report
Comments and Suggestions for Authors
1. The names of 10 SSA countries should be included in the abstract.
2. Keywords should be listed in alphabetical order for clarity. In keywords, the use of words used in the title should be avoided.
3. In the text, the WHO malaria 2023 report is referred to in 2 places with different numbers. References 1 and 27 both refer to the same report.
4. The references within the text and in the reference section need to be formatted according to the journal's style guide (e.g., publication years, punctuation after journal titles, italicization of journal names, correction of abbreviation errors, and proper formatting of species names). et al. etc.
5. I suggest that a map showing the African countries used in the research on the continent should be used in the text.
6. All species names should be written out in full the first time they are mentioned. Subsequently, abbreviations should be used throughout the text. For example, see Line 95 (P. falciparum,' etc.).
7. All species names should be italic in references section.
Abbreviations are not used in the text in a certain standard. All abbreviations should be checked. If something is to be abbreviated, it should first be given exactly where it is first used and then the abbreviation should be used continuously. e.g. SSA and ITNs
There are too many tables and numbers in the text. this is very distracting for the readers. Maybe some simpler tables or graphical data can be given for comparisons?
Comments on the Quality of English LanguageMinor editing
Round 2
Reviewer 3 Report
Comments and Suggestions for Authors
Check the formatting of journal names in the references
Comments on the Quality of English Language
Good